# SIMPLE PARAMETER-FREE SELF-ATTENTION APPROXIMATION

**Yuwen Zhai**[1*]  **Jing Hao**[2*]  **Liang Gao**[1]  **Xinyu Li**[1†]  **Yiping Gao**[1]  **Shumin Han**[2]
[1]Huazhong University of Science and Technology   [2]Baidu VIS
{yuwenzhai, gaoliang, lixinyu, gaoyiping}@hust.edu.cn
{haojing08, hanshumin}@baidu.com

## ABSTRACT

The hybrid model of self-attention and convolution is one of the methods to lighten ViT. The quadratic computational complexity of self-attention with respect to token length limits the efficiency of ViT on edge devices. We propose a self-attention approximation without training parameters, called SPSA, which captures global spatial features with linear complexity. To verify the effectiveness of SPSA combined with convolution, we conduct extensive experiments on image classification and object detection tasks. The source code is available at SPSA.

## 1 INTRODUCTION

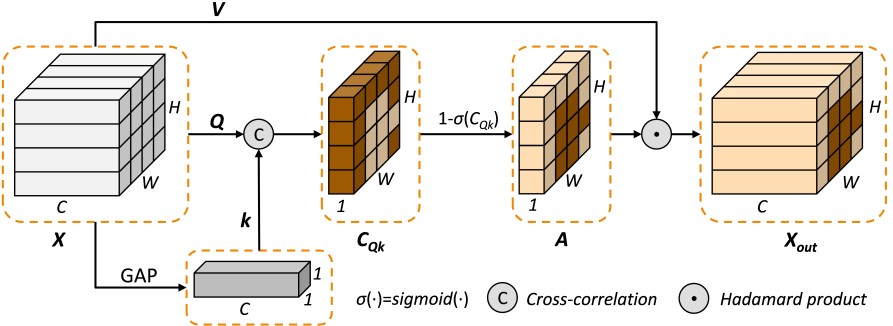

Figure 1: Illustration of simple parameter-free self-attention approximation.

Recently, hybrid models of convolution and self-attention have emerged as an important approach to apply ViT (Kolesnikov et al., 2021) to edge devices (Li et al., 2022; Maaz et al., 2022). We propose a parameter-free self-attention approximation, called SPSA, which is extremely simple and has linear computational complexity. The paper aims to present a new form of global spatial attention that inspires researchers to design new lightweight CNN-ViT hybrid networks. Based on this, we experimentally validate the feasibility of combining SPSA with convolution taking the well-known channel attention as the performance benchmark. To demonstrate the effectiveness of SPSA as spatial attention, we explore the generalizability of the fusion of SPSA and different channel attention.

## 2 METHOD

Fig. 1 illustrates SPSA. Given an input $X \in \mathbb{R}^{H \times W \times C}$, we implement global average pooling (GAP) to obtain the key vector $k$, i.e., $k = \frac{1}{WH} \sum_{i=1,j=1}^{W,H} X_{ij}$ and $k \in \mathbb{R}^C$. The matrices $Q$ and $V$ are generated using identical mappings, i.e., $Q = V = X$. According to Eq. (1), SPSA uses the cross-correlation coefficient to evaluate the similarity between each location in query $Q$ and key $k$.

$$C_{Qk} = \frac{\sum_{i=1}^{C} [Q_{:,:,i} - \bar{Q}][k_i - \bar{k}]}{\sqrt{\sum_{i=1}^{C} [Q_{:,:,i} - \bar{Q}]^2 \sum_{i=1}^{C} [k_i - \bar{k}]^2}}, \tag{1}$$

The weight matrix $A$ is get by subtracting the normalized $C_{Qk}$ from 1 (reverse operation), i.e, $A = (1 - \sigma(C_{Qk}))^\alpha$, where $\sigma(\cdot)$ is a Sigmoid function, the exponent $\alpha$ is used to enhance feature

---

*Equal contribution.  †Corresponding author.

expression. Finally, $A \in \mathbb{R}^{W \times H \times 1}$ is expanded along the channel to the size of $V$, and the output $X_{out}$ is the Hadamard product of $A$ and $V$, i.e., $X_{out} = \text{expand}(A) \circ V$. Inspired by self-attention, multi-head SPSA is calculated in sub-channels to enhance the expression of feature subspaces.

Obviously, SPSA has a linear complexity $O(N)$ to the number of pixels and has no learnable parameters. Importantly, SPSA gives different attention to the global spatial content and is competent for the approximation of self-attention. If you are interested, we explain the code, the cross-correlation coefficient, the reverse operation and the computational complexity in detail in the appendix.

Table 1: Classification results on ImageNet-1k.

| Method | + Param. | FLOPs | Inference | Top-1 |
|---|---|---|---|---|
| ResNet-50 | **0** | **4.11G** | **1879** | 77.28 |
| + SE | 2.51M | 4.12G | 1510 | 77.86 |
| + CBAM | 2.51M | 4.12G | 1286 | 78.24 |
| + GSoPNet1 | 2.73M | 6.39G | 1359 | **79.01** |
| + AANet | 0.24M | 4.15G | - | 77.70 |
| + ECA | 80 | 4.12G | 1769 | 77.99 |
| + FCA | 2.51M | 4.12G | 1453 | 78.57 |
| + SPSA | **0** | **4.11G** | 1644 | 78.08 |
| + SPSA-SE | 2.51M | 4.12G | 1410 | 78.31 |
| + SPSA-ECA | 80 | 4.12G | 1442 | 78.25 |
| + SPSA-FCA | 2.51M | 4.12G | 1256 | 78.69 |
| ResNet-101 | **0** | **7.83G** | **1129** | 78.72 |
| + SE | 4.74M | 7.85G | 960 | 79.19 |
| + AANet | 0.85M | 8.05G | - | 78.70 |
| + ECA | 165 | 7.84G | 1003 | 79.09 |
| + FCA | 4.74M | 7.85G | 933 | 79.63 |
| + SPSA | **0** | **7.83G** | 968 | 79.42 |
| + SPSA-SE | 4.74M | 7.85G | 896 | 79.60 |
| + SPSA-ECA | 165 | 7.84G | 934 | 79.49 |
| + SPSA-FCA | 4.74M | 7.85G | 808 | **79.65** |
| ResNet-152 | **0** | **11.56G** | **805** | 79.39 |
| + SE | 6.58M | 11.58G | 758 | 79.84 |
| + AANet | 1.41M | 11.90G | - | 79.10 |
| + ECA | 250 | 11.57G | 785 | 79.86 |
| + FCA | 6.58M | 11.58G | 713 | **80.02** |
| + SPSA | **0** | **11.56G** | 764 | 79.99 |

Table 3: SPSA application on other baselines.

| Baselines | Top-1 | Top-5 | Mothod | Top-1 | Top-5 |
|---|---|---|---|---|---|
| ResNeXt-50 | 78.35 | 94.11 | + SPSA | **78.89** | **94.47** |
| MobileNetV2 | 67.09 | 87.92 | + SPSA | **67.89** | **88.40** |
| ShuffleNetV2 | 65.45 | 86.54 | + SPSA | **65.87** | **86.72** |

Table 2: Object detection results on COCO 2017.

| Method | + Param. | FLOPs | AP | $AP_{50}$ | $AP_{75}$ |
|---|---|---|---|---|---|
| Faster-RCNN | | | | | |
| ResNet-50 | **0** | **207.07** | 36.4 | 58.2 | 39.2 |
| + SE | 2.51M | 207.18 | 37.7 | 60.1 | 40.9 |
| + ECA | 80 | 207.18 | 38.0 | 60.6 | 40.9 |
| + FCA | 2.51M | 207.18 | 39.0 | 61.1 | 42.3 |
| + SPSA | **0** | **207.07** | 39.0 | 60.3 | 42.2 |
| + SPSA-SE | 2.51M | 207.18 | **39.5** | **61.2** | **42.9** |
| + SPSA-ECA | 80 | 207.18 | 39.3 | **61.2** | 42.6 |
| + SPSA-FCA | 2.51M | 207.18 | 39.4 | 61.0 | 42.6 |
| ResNet-101 | **0** | **283.14** | 38.7 | 60.6 | 41.9 |
| + SE | 4.74M | 283.33 | 39.6 | 62.0 | 43.1 |
| + ECA | 165 | 283.32 | 40.3 | **62.9** | 44.0 |
| + FCA | 4.74M | 283.33 | 41.2 | **63.3** | 44.6 |
| + SPSA | **0** | **283.14** | 41.2 | 62.5 | 45.0 |
| + SPSA-SE | 4.74M | 283.33 | 41.3 | 62.8 | 45.2 |
| + SPSA-ECA | 165 | 283.32 | **41.6** | 62.7 | **45.3** |
| + SPSA-FCA | 4.74M | 283.33 | 41.5 | 62.8 | 45.2 |
| Mask-RCNN | | | | | |
| ResNet-50 | **0** | **260.14** | 37.2 | 58.9 | 40.3 |
| + SE | 2.51M | 260.25 | 38.7 | 60.9 | 42.1 |
| + 1NL | 8.40M | 268.54 | 39.0 | 61.1 | 41.9 |
| + ECA | 80 | 260.25 | 39.0 | 61.3 | 42.1 |
| + FCA | 2.51M | 260.25 | 40.3 | **62.0** | 44.1 |
| + SPSA | **0** | **260.14** | 39.5 | 60.5 | 43.1 |
| + SPSA-SE | 2.51M | 260.25 | **40.5** | **61.6** | **44.2** |
| + SPSA-ECA | 80 | 260.25 | 40.0 | 61.5 | 43.6 |
| + SPSA-FCA | 2.51M | 260.25 | 40.4 | 61.7 | 44.0 |
| RetinaNet | | | | | |
| ResNet-50 | **0** | **239.32** | 35.6 | 55.5 | 38.2 |
| + SE | 2.51M | 239.43 | 37.1 | 57.2 | 39.9 |
| + ECA | 80 | 239.43 | 37.3 | 57.7 | 39.6 |
| + SPSA | **0** | **239.32** | 37.5 | 56.9 | 39.9 |
| + SPSA-SE | 2.51M | 239.43 | **38.6** | **58.0** | **41.2** |
| + SPSA-ECA | 80 | 239.43 | 38.2 | 57.8 | 40.6 |

## 3 EXPERIMENTS & CONCLUSION

**Image classification** On ImageNet-1k, we carry out ResNet-SPSA experiments and take other modules as performance benchmarks, including SE (Hu et al., 2018), CBAM (Woo et al., 2018), GSoP-Net1 (Gao et al., 2019), AANet (Bello et al., 2019), ECA (Wang et al., 2020), and FCA (Qin et al., 2021). As in Table 1, SPSA achieves a better trade-off in accuracy-parameter and accuracy-inference speed. Admittedly, there is a gap between SPSA and SOTA method, but it decreases as the baseline increases. SPSA fused with other channel attentions also all achieve better performance, except for FcaNet which is incompatible with SPSA due to DCT. The results are sufficient to demonstrate the feasibility of SPSA combined with convolution and its effectiveness as spatial attention. Table 3 further indicates the generalizability of SPSA to other types of convolutions.

**Object Detection** In Table 2, we evaluate SPSA using Faster-RCNN (Ren et al., 2015), Mask-RCNN (He et al., 2017) and RetinaNet (Lin et al., 2017) as detectors and ResNets with FPN as the backbone. Surprisingly, SPSA achieves almost the same performance as the SOTA method. Notably, SPSA surpasses NL (Wang et al., 2018), which is also a form of self-attention.

Extensive experiments have demonstrated the feasibility of SPSA as a self-attentive approximation. We trust that this new form of self-attention will have potential in lightweight CNN-ViT hybrid models and inspire researchers to apply it to new model designs.

URM Statement

Author Yuwen Zhai meets the URM criteria of ICLR 2023 Tiny Papers Track.

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

## A  Appendix

### A.1  Code of SPSA

SPSA module is extremely simple to implement. As in Figure 2, we give a reference implementation of SPSA in PyTorch. Multi-head SPSA simply adds one dimension to the input and adjusts the dimension index of the calculation.

```python
def SPSA_Attention(x, alpha):
    # x: input feature with shape [N,C,H,W]
    # alpha: exponent
    k = x.mean(dim=[-1, -2])  # N,C,1,1
    kd = torch.sqrt((k - k.mean(dim=1)).pow(2).sum(dim=1))  # N,1,1,1
    Qd = torch.sqrt((x - x.mean(dim=1)).pow(2).sum(dim=1))  # N,1,H,W

    # cross-correlation coefficient matrix C_Qk
    C_Qk = (((x - x.mean(dim=1)) * (k - k.mean(dim=1))).sum(dim=1)) / (Qd * kd)  # N,1,H,W

    # weight matrix
    A = (1 - sigmoid(C_Qk)) ** alpha  # N,1,H,W

    # Hadamard product
    out = x * A  # N,C,H,W
    return out
```

Figure 2: PyTorch code of the proposed SPSA module

### A.2  Implementation Details

#### A.2.1  ImageNet-1k

Recall that we compare SPSA with other methods on ImageNet-1k taking ResNet (He et al., 2016) families as the backbones. We also apply SPSA to MobileNetV2 (Sandler et al., 2018), ShuffleNetV2 (Ma et al., 2018) and ResNeXt (Xie et al., 2017) to verify its generalization. For all backbone networks, we employ exactly the same data augmentation and hyperparameter settings as in (He et al., 2016) and (Hu et al., 2018). Specifically, the input images are randomly cropped to 224×224 with random horizontal flipping. We use an SGD optimizer with a momentum of 0.9 and a weight decay of 1e-4. The initial learning rate is set to 0.1 for a batch size of 256 (using 4 GPUs with 64 images per GPU) with the linear scaling rule (Goyal et al., 2017) and a linear warm-up of 5 epochs. All models are trained within 100 epochs with cosine learning rate decay and label smoothing following FcaNet (Qin et al., 2021). We use the Nvidia APEX mixed precision training toolkit

for training efficiency. For the testing on the validation set, the shorter side of an input image is first resized to 256, and a center crop of $224 \times 224$ is used for evaluation.

### A.2.2 MS COCO

We use MMDetection toolkit (Chen et al., 2019) for experiments on MS COCO dataset with the pre-trained ResNet-50 and ResNet-101 as the backbones for the detector. Specifically, the shorter side of the input image is resized to 800. The SGD optimizer has a weight decay of 1e-4, a momentum of 0.9, and a batch size of 8 (4 GPUs with two images per GPU) within 12 epochs. The learning rate is initialized to 0.01 and is decreased by the factor of 10 at the 8th and 11th epochs, respectively. In validation, we report the standard Average Precision (AP) under IOU thresholds ranging from 0.5 to 0.95 in increments of 0.05. We also retain AP scores for small, medium and large objects.

### A.3 DISCUSSION OF THE CROSS-CORRELATION COEFFICIENT AND THE COSINE-SIMILARITY

In this subsection we review and discuss the cross-correlation coefficient and the cosine-similarity. Given two sets of vectors $x \in \mathbb{R}^N$ and $y \in \mathbb{R}^N$.

**Cross-correlation coefficient** The population cross-correlation coefficient $\rho_{x,y}$ is defined as the quotient of the covariance and standard deviation between the two variables.

$$\rho_{x,y} = \frac{\text{cov}(x,y)}{\sigma_x \sigma_y} = \frac{E[(x - \mu_x)(y - \mu_y)]}{\sigma_x \sigma_y}, \tag{2}$$

where $\text{cov}(x,y)$ is the covariance of $x$ and $y$, and $\sigma_x$, $\sigma_y$ are the standard deviations of $x$ and $y$, respectively. Estimating the covariance and standard deviation of the samples, the sample cross-correlation coefficient $C_{x,y}$ is obtained as:

$$C_{x,y} = \frac{\sum_{i=1}^n (x_i - \bar{x})(y - \bar{y})}{\sqrt{\sum_{i=1}^n (x_i - \bar{x})^2 \sum_{i=1}^n (y - \bar{y})^2}} \tag{3}$$

where $\bar{x} = \frac{1}{n}\sum_{i=1}^n x_i$, $\bar{y} = \frac{1}{n}\sum_{i=1}^n y_i$. In this paper, we use the above equation to evaluate the pairwise affinity between pixels.

**Cosine-similarity** According to Euclid's dot product formula

$$x \cdot y = \|x\| \|y\| \cos \theta, \tag{4}$$

the cosine-similarity $Cos_{x,y}$ between the two vectors is obtained

$$Cos_{x,y} = \cos \theta = \frac{x \cdot y}{\|x\| \|y\|} = \frac{\sum_{i=1}^n x_i y_i}{\sqrt{\sum_{i=1}^n x_i^2 \sum_{i=1}^n y_i^2}}. \tag{5}$$

Comparing Eq. (3) and Eq. (5) to obtain Eq. (6), it shows that the cross-correlation coefficient is the cosine-similarity after the data centering process. Therefore, the cross-correlation coefficient is less sensitive to fluctuations in the data than the cosine-similarity.

$$C_{x,y} = \frac{\sum_{i=1}^n (x_i - \bar{x})(y - \bar{y})}{\sqrt{\sum_{i=1}^n (x_i - \bar{x})^2 \sum_{i=1}^n (y - \bar{y})^2}} = \frac{(x - \bar{x}) \cdot (y - \bar{y})}{\|x - \bar{x}\| \|y - \bar{y}\|} = Cos_{x-\bar{x}, y-\bar{y}} \tag{6}$$

Since $\|x\| \|y\|$ in Eq. (5) would complicate the computation, $\frac{x \cdot y}{\sqrt{d}}$ is used as an alternative in self-attention mechanism to evaluate the similarity between paired vectors, where $d$ points to the vectors' dimensions. Eq. (4) shows that for larger values of $d$, the larger dot product's magnitude will affect the similarity representation and push the softmax function to the regions with extremely small gradients (Vaswani et al., 2017). To counteract this effect, self-attention scale the dot product by $\frac{1}{\sqrt{d}}$. The dot product in self-attention is implemented by highly optimized matrix multiplication code to achieve high parallelism. In contrast, in SPSA architecture, each position in $Q$ is only required to match the similarity with a single $k$ vector, which has high parallelism. Therefore, the cross-correlation coefficient with low sensitivity to data is allowed to be applied as an indicator of pairwise affinity.

In addition, we experimentally verified the superiority of the cross-correlation coefficient over the dot product cosine-similarity in SPSA architecture. We experiment on ImageNet-1k and Mini-ImageNet datasets by replacing the cross-correlation coefficient with dot product. Table 4 demonstrates that dot product similarity does not work well in SPSA. It shows that using the dot product to calculate the similarity to assess the affinity between $Q$ and vector $k$ is insufficient. The cross-correlation coefficient is a better choice.

Table 4: Comparison experiments of different evaluation methods with ResNet50 as baseline.

| Method | ImageNet-1k (Top-1) | Mini-ImageNet (Top-1) |
|---|---|---|
| Baseline | 77.28 | 80.55 |
| Dot product cosine-similarity | 75.69 | 80.32 |
| Cross-correlation coefficient | 78.08 | 81.59 |

### A.4 ANALYSIS OF COMPUTATIONAL COMPLEXITY

Unlike self-attention, query $Q$ and value $V$ of SPSA are obtained utilizing an identical mapping of $X$, i.e., $O_Q = O_V = 0$. The computational complexity of $k$ vector obtained by GAP is

$$O_k = O(HWC). \tag{7}$$

We estimate the correlation coefficient matrix Eq. (1) to obtain the computational complexity of generating the weight $A$

$$O_{\text{Cross}} = O\left(HW\left(C + C^2\right)\right). \tag{8}$$

The computational complexity of acting $A$ on $V$ via the Hadamard product is

$$O_{\text{Act}} = O(HWC). \tag{9}$$

Thus, the overall computational complexity of SPSA is

$$O_{\text{SPSA}} = O_k + O_{\text{Cross}} + O_{\text{Act}} = O\left(3HWC + HWC^2\right). \tag{10}$$

Compared with self-attention, SPSA has linear complexity for the number of pixels.

### A.5 EXPLANATION OF REVERSE OPERATION

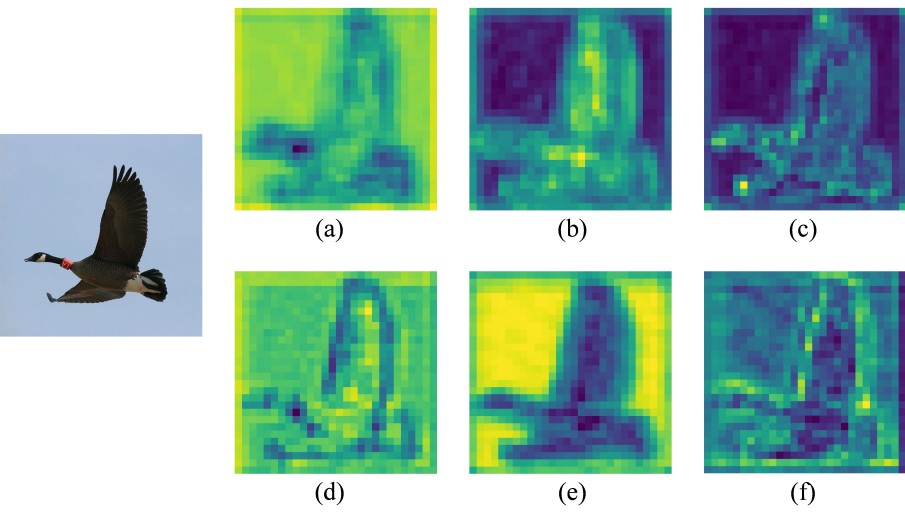

Figure 3: ResNet-50 visualization of SPSA module at layer2.3. (a)-(c) with reverse and (d)-(f) without reverse. (a)(d), (b)(e), and (c)(f), each group represents the input, the attention weight, and the output of SPSA, respectively.

As described in section 2, SPSA gets the key vector $k$ by the feature map's global average pooling (GAP). It causes SPSA to work differently than the intuition that comes from self-attention. Specifically, GAP is challenging to capture the complex information in the feature maps and misses most of the detailed features (Qin et al., 2021). In contrast to the general features in the global scope represented by GAP, we believe that spatial attention should enhance special features, such as texture details. Intuitively, enhancing special detail features is helpful for visual recognition tasks. Therefore, we use the reverse operation to enhance the specificity features rather than features similar to the $k$ vector generated by GAP.

To intuitively discuss the necessity of the reverse operation in the system, the feature map of SPSA module is visualized in Fig. 3. Fig. 3 (a) and (d) show the inputs of SPSA module in layer 2.3. Both are generally similar, and with the network optimized iteratively, the feature maps have the

same attention to the target and the background. Fig. 3 (b) and (e) show the attention weights obtained from the cross-correlation calculation in SPSA, which have opposite results. The reverse operation drives SPSA to focus almost on the object itself, while SPSA without reverse focuses almost exclusively on the background region. It proves that the reverse operation directly affects the region of attention of SPSA. Naturally, in Fig. 3 (c) and (f), the final outputs show that the SPSA without reverse tends to focus on the background. The reversed SPSA drives the network to focus on the object, which is more beneficial for visual tasks.

