# OpenReview forum: "Simple Parameter-free Self-attention Approximation"
_ICLR.cc/2023/TinyPapers — Submitted to Tiny Papers @ ICLR 2023_

### Official Review · Reviewer_xHf8 · 2023-03-30

**Confidence:** 4

**Summary Of Contributions:**

This paper aims to propose a self-attention approximation (SPSA) that does not require any parameters and operates at linear complexity with respect to the number of pixels. The authors demonstrate comparable performance against several methods in the image classification and detection task. Additional visualizations and ablations verify the design choices.

**Rating:**

High Potential (HP): a submission which meets the reviewing criteria and has potential to make an impact on the field

**Strengths And Weaknesses:**

**Clarity**

The paper is well-written and easy to follow. Implementation details and experiments are well-outlined. While reading the main text, I found the design choice for reversing the $C_{qk}$ matrix to obtain $A$ a bit confusing (which was later discussed in the Appendix) and I would highly recommend the authors include an extra line to explain their motivation in the main text for better clarity. A minor error, Sec. 2: "The weight matrix $A$ is get by "

**Correctness**

The authors initially motivate their method by discussing the computational complexity of ViTs that limits them from being deployed on edge devices and therefore design a new lightweight CNN-ViT hybrid network. This seems to be a bit misleading as SPSA is compared against methods that leverage attention as a feature enhancement strategy (e.g channel attention) as opposed to being used as a primary building block (e.g the hybrid ViT discussed in [1], [2]).

- *[1] AN IMAGE IS WORTH 16X16 WORDS: TRANSFORMERS FOR IMAGE RECOGNITION AT SCALE Kolesnikov et. al '21*
- *[2] HOW DO VISION TRANSFORMERS WORK? Park et. al '22*

**Questions**

I (sort of) understand the notion of reversing the $C_{qk}$ matrix, but it still seems a bit counterintuitive to me. The global average pooled vector should represent the overall content present in the image (which in this case should indicate the object present in the image) and therefore it's surprising to see the key vector have maximum similarity with the background of the image (Fig. 3(e)). I would highly appreciate it if the authors could provide some clarity in this regard.

The authors compare the cross-correlation coefficient against cosine similarity and argue that the cross-correlation matrix is effectively the cosine similarity operation after data centering. I am curious, is there any normalization (e.g layer normalization) applied after the block and if it is, would the cosine similarity + layer norm lead to similar results?

**Suggested Changes:**

Overall, I appreciate the proposed strategy due to its simplicity and effectiveness. I strongly believe it will motivate further work in the area. However, I would sincerely appreciate a few changes:
- The authors must clarify what they imply by a "hybrid CNN-ViT network". Methods like ResNet + SE, ECA, SE, etc are not commonly referred to as CNN-ViT networks.
- I would also appreciate a discussion involving SPSA in a pure-attention network. It probably does not make sense to propose a pure SPSA network, but rather replace a few Multi-Head Self Attention (MHSA) blocks used in ViTs with SPSA and compare their performance. This could provide an idea of how close SPSA models the more expensive MHSA operation.

---

### Meta-Review · Area_Chair_RBCr · 2023-04-08

**Recommendation:** Invite to present
**Confidence:** 4

**Metareview:**

The paper proposes a self-attention approximation method called SPSA that operates at linear complexity and does not require any parameters. The proposed method is demonstrated to achieve comparable performance against several methods in the image classification and detection task, and additional visualizations and ablations verify the design choices.

Reviewers praise the clarity of the paper and its well-outlined implementation details and experiments. However, the authors are encouraged to clarify the design choice of reversing the matrix in the main text and provide additional discussions regarding SPSA in a pure-attention network and what they imply by a "hybrid CNN-ViT network". Reproducibility is also a concern, as the authors did not provide the implementation code. All the sources should be released upon acceptance.

**Summary:**

The paper proposes a self-attention approximation method, SPSA, for image classification and detection tasks. The paper is praised for its clarity and well-outlined experiments.

**Reason For Not Giving A Higher Recommendation:**

Despite overall good clarity, the method could be better motivated and presented. The lack of implementation code hinders the reproducibility of the results, and the authors are encouraged to release the code. Additionally, the design choice of reversing the matrix could be clarified in the main text. Ablation studies are encouraged to understand the reverse operation in SPSA.

**Reason For Not Giving A Lower Recommendation:**

The paper proposes a novel self-attention approximation method called SPSA that operates at linear complexity with respect to the number of pixels, demonstrating comparable performance against several methods in the image classification and detection task. The paper is well-written, and the experiments are well-outlined, making it a strong candidate for publication. While the lack of implementation code and the need for improved motivation and presentation prevent a higher rating, the overall quality and potential impact of this work warrant an invitation to present.

---

### Decision · Program_Chairs · 2023-04-09

Invite to present